# Thermal Niche for Seed Germination and Species Distribution Modelling of *Swietenia macrophylla* King (Mahogany) under Climate Change Scenarios

**DOI:** 10.3390/plants10112377

**Published:** 2021-11-04

**Authors:** Salvador Sampayo-Maldonado, Cesar A. Ordoñez-Salanueva, Efisio Mattana, Michael Way, Elena Castillo-Lorenzo, Patricia D. Dávila-Aranda, Rafael Lira-Saade, Oswaldo Téllez-Valdés, Norma I. Rodriguez-Arevalo, Tiziana Ulian, Cesar M. Flores-Ortíz

**Affiliations:** 1Plant Physiology Laboratory, Biotechnology and Prototypes Unit (UBIPRO), FES Iztacala, Universidad Nacional Autónoma de México (UNAM), Tlalnepantla 54090, Estado de Mexico, Mexico; ssampayom@hotmail.com (S.S.-M.); caos@unam.mx (C.A.O.-S.); 2Royal Botanic Gardens, Kew, Wakehurst, Ardingly, Haywards Heath RH17 6TN, West Sussex, UK; e.mattana@kew.org (E.M.); m.way@kew.org (M.W.); E.CastilloLorenzo@kew.org (E.C.-L.); t.ulian@kew.org (T.U.); 3Natural Resources Laboratory, Biotechnology and Prototypes Unit (UBIPRO), FES Iztacala, Universidad Nacional Autónoma de México (UNAM), Tlalnepantla 54090, Estado de Mexico, Mexico; pdavilaa@unam.mx (P.D.D.-A.); rlira@unam.mx (R.L.-S.); tellez@unam.mx (O.T.-V.); 4Seed Bank, FES Iztacala, Universidad Nacional Autónoma de México (UNAM), Tlalnepantla 54090, Estado de Mexico, Mexico; isela.unam@gmail.com; 5National Laboratory in Health, FES Iztacala, Universidad Nacional Autónoma de México (UNAM), Tlalnepantla 54090, Estado de Mexico, Mexico

**Keywords:** mahogany tree, seed germination, cardinal temperatures, thermal time, climate change, potential distribution

## Abstract

*Swietenia macrophylla* is an economically important tree species propagated by seeds that lose their viability in a short time, making seed germination a key stage for the species recruitment. The objective of this study was to determine the cardinal temperatures and thermal time for seed germination of *S. macrophylla*; and its potential distribution under different climate change scenarios. Seeds were placed in germination chambers at constant temperatures from 5 to 45 °C and their thermal responses modelled using a thermal time approach. In addition, the potential biogeographic distribution was projected according to the Community Climate System Model version 4 (CCSM4). Germination rate reached its maximum at 37.3 ± 1.3 °C (To); seed germination decreased to near zero at 52.7 ± 2.2 °C (ceiling temperature, Tc) and at 12.8 ± 2.4 °C (base temperature, Tb). The suboptimal thermal time $\left(\theta_{1}\left(50\right)\right)$ needed for 50% germination was ca. 190 °Cd, which in the current scenario is accumulated in 20 days. The CCSM4 model estimates an increase of the potential distribution of the species of 12.3 to 18.3% compared to the current scenario. The temperature had an important effect on the physiological processes of the seeds. With the increase in temperature, the thermal needs for germination are completed in less time, so the species will not be affected in its distribution. Although the distribution of the species may not be affected, it is crucial to generate sustainable management strategies to ensure its long-term conservation.

## 1. Introduction

*Swietenia macrophylla* King (Meliaceae), commonly known as mahogany, is one of the most important timber resources in the Americas and the most valuable in Mexico [1,2]. Physical, chemical, mechanical, and anatomical properties of its wood contribute to its high economic value. It is in high demand for production of plywood and boards for the international markets in the United States, Japan, and Europe [3]. It is used for the construction of homes, boats, and fine cabinetmaking. In addition, limonoids, bioactive compounds in its leaves, seeds, and bark [4] are used in medicinal [5,6] and antifungal products [7].

According to Negreros-Castillo et al. [8], *S. macrophylla* is considered a “financial keystone species” because its absence increases the probability that the forest will be converted to other lucrative land uses. However, there is a need to incentivize its protection and sustainable management in the tropical forest. It is therefore highly important to generate information that helps understand the physiological processes involved in the seed germination as a criterion for the regeneration and conservation of the species.

*S. macrophylla* is a deciduous tree native to the Americas, distributed from Mexico along the Atlantic coast of Central America to the northern Amazon among Brazil, Colombia, Peru, and Bolivia [9,10]. Trees dominate the upper canopy of the Mexican Tropical Forest [11]. According to Pennington and Sarukhan [12], it is distributed along the Gulf of Mexico watershed, with populations found from southern Tamaulipas to northern Puebla, Veracruz, and the Yucatan Peninsula.

Given the high quality of its wood, this species has been overexploited, and threats to this species are exacerbated by its vulnerability to pest [13]. Therefore, the distribution of this species is fragmented in small, scattered populations, and is affected by a shrinking gene pool [14]. According to Newton [15] and IUCN [16], it is considered vulnerable to extinction, and was therefore included in 2003 in Appendix II of the Convention on International Trade in Endangered Species of Wild Fauna and Flora [17]. Similarly, Martins-Leão et al. [2], mention that in Brazil and other regions it is in danger of extinction. In addition, it has been reported that the seeds of this species are sensitive to desiccation [18], as they cannot support the extreme loss of cellular water and quickly lose their viability. Therefore, it is necessary to develop suitable techniques for the long-term storage of seeds sensitive to desiccation.

The tropical forest supports the greatest diversity in the country, which is why it is the basis for the development of the forest regions of Mexico. However, it is a fragile ecosystem and highly vulnerable to climate change [19]. The mechanism for its regeneration and ecological restoration is through seed production, but its success depends on the ability of those seeds to germinate [20] and withstand climate change which is threatening the ecosystem function and diversity of forest regions [21].

According to Calvo et al. [22], by the end of the last century, more than 60% of the original distribution of the tropical forest in the Americas had been lost. In the case of Mexico, only 24% of its natural populations have been conserved. Garza-López et al. [14] projected that in 2030, under a scenario of intermediate concentrations of CO_2_ (RCP6.0 W/m^2^), there would be a 60% reduction in the distribution in the climatic habitat of *S. macrophylla* in the Yucatan Peninsula. However, Navarro-Martínez et al. [9] mention that the populations of the species are stable and abundant due to community forest management in the peninsula.

The tree produces fruits in the form of ovoid dehiscent capsules that mature from February to March, which then disperse seeds (samaras) from March to April. This coincides with the loss of leaves, allowing the free circulation of wind which favors dispersal up to 50 m away [23]. According to Dutta et al. [6], the seeds are not toxic and are used in traditional medicine [4,24]. In the literature, the seeds are classified as intermediate [18] or recalcitrant seeds (desiccation sensitive) [25,26]. This can be explained by the fact that the species has a wide distribution range, and each population has specific environmental conditions. Thus, it is important to know the origin of the collections. Seeds quickly lose their viability (short-lived seeds), but when freshly harvested they can reach germination percentages up to 95% [26]. Special conditions are therefore required for their storage and conservation. The populations where the collections were made for this research are located in Zozocolco de Hidalgo, in the state of Veracruz, the seeds presented intermediate characteristics.

Germination is the most critical stage of plant development [27], and temperature is one of the most important environmental factors, regulating the speed and maximum percentage of germination [28]. Previous studies have reported *S. macrophylla* seeds reached higher germination values when incubated at warm temperatures [29,30]. However, in reviewing the literature, to our knowledge, no prior study has reported the cardinal temperatures or thermal time for seed germination of *S. macrophylla* seeds. Therefore, it is important to carry out studies of the effect of temperature on the physiological processes of germination to find the thresholds of the species [31], to determine the optimal temperature at which the highest percentage of germination occurs in the shortest time [32]; and the accumulated temperature (thermal time), necessary to reach a certain development process or phenological stage [33]. The aim of this study was to determine the cardinal temperatures and the thermal time for seed germination of *S. macrophylla*, as well as to analyze the impact of future climate change scenarios on its germination and potential biogeographic distribution.

## 2. Results

### 2.1. Germination

The average germination over the temperature range was above 70% and ranged from 4 to 96% (Figure 1), with these differences being statistically significant (F_6,28_ 54.02; *p* < 0.00001). The lowest germination percentage was observed at 15 °C, showing an abrupt decrease. It is important to note that at low temperatures (5 and 10 °C) there was no germination (Figure 1). The highest germination rate was at 25 °C and there was a tendency to decrease germination rate with increasing temperature, though there was no significant difference among germination rates recorded between 20 and 35 °C.

At temperatures from 30 to 45 ± 2 °C, seeds of *S. macrophylla* began to germinate after 6 days, while at temperatures of 20 and 25 ± 2 °C this occurred after 18 days. In the case of the 15 ± 2 °C temperature treatment, this was after 91 days (Figure 2). The time to germination of 50% of the seeds showed significant differences among temperatures (F_5,24_ 2.51; *p* < 0.02). At 35 ± 2 °C, 50% germination occurred at 9 days. At higher temperatures, less time was required than at lower temperatures; for example, at 40 and 45 ± 2 °C, 50% germination was achieved in 10 d and 17 h and 10 days and 3 h, respectively. Meanwhile, at 30 ± 2 °C this occurred in 19 days, while more than 36 days were required at 25 ± 2 and 20 ± 2 °C (Figure 2).

We observed significant differences in germination speed among treatments (GS; F_6,28_ 32.61; *p* < 0.00001). The fastest germination speed occurred at 35 ± 2 °C, in which one seed germinated per day. As temperature increased, germination speed decreased, and at 45 ± 2 °C one seed germinated every two days. At temperatures below 35 ± 2 °C, germination slowed, requiring 100 days for a seed to germinate at 15 ± 2 °C.

### 2.2. Cardinal Temperatures

Figure 3 shows the germination rate values for each temperature of the different percent fractions in the seed lot. The suboptimal temperatures for germination speed of this species were explained by the model with 86%, while the supra-optimal temperatures were 91% explained. The calculation of the intercept of the slopes for each percentile indicated that the Tb (base temperature) for germination to occur was 12.8 ± 2.4 °C. As the temperature increased, germination speed increased until reaching its highest at a To (optimal temperature) of 37.3 ± 1.3 °C. When the temperature increased above To, the germination speed decreased until a Tc (ceiling temperature) of 52.7 ± 2.2 °C.

As expected, the Tb of 12.8 ± 2.4 was below 15 ± 2. °C since the average germination rate was only 4% at this temperature after 100 days (Figure 1). The To (37.3 ± 1.3 °C) is directly related to the temperature that presented the fastest germination speed, while Tc was 52.7 ± 2.2 °C, and can be explained because at 45 ± 2 °C only 56% of the seeds germinated and they required 10 days and 3 h for 50% of the seed lot to germinate (Figure 2).

### 2.3. Thermal Time

The thermal time for 50% germination of the seed lot for the sub-optimal temperatures $\left(\theta_{1}\left(50\right)\right)$ was 189.091 ± 4.239 °Cd. The Probit model explained this with more than 92% accuracy (Table 1). The thermal time for 50% germination of the seed lot in the supra-optimal temperature range $\left(\theta_{2}\left(50\right)\right)$ was 119.613 ± 3.721 °Cd, the Probit model explained this with more than 96%.

The thermal time in the sub-optimal range in Figure 4a shows that as more heat is accumulated, the probability of obtaining higher germination percentages increases because it approaches a thermal time $\left(\theta_{1}\left(50\right)\right)$ of 189.091 ± 4.239 °Cd, at which 50% of the seed lot germinates. In the case of thermal time in the supra-optimal range, as heat accumulates, the germination percentages decrease, which reflects the difference in thermal time $\left(\theta_{2}\left(50\right)\right)$ of 119.613 ± 3.721 °Cd necessary for germination of 50% of the seed lot (Figure 4b).

### 2.4. Germination under Climate Change Scenarios

The seed dispersal month for *S. macrophylla* is March which has an average temperature of 22 °C. For the intermediate future (2050), under the conservative scenario (RCP4.5 Watts/m^2^), the French model (CNRMCM5) predicts an increase in temperature of 2.3 °C for that month, while the American (GFDL-CM3) model predicts 3.2 °C, the English model (HADGEM2-ES) 2.1 °C, and the German model (MPI-ESM-LR) 2.6 °C.

In the current scenario, the thermal time ($\theta_{1}\;\left(50\right))$ is accumulated in 20.5 days. For the intermediate future (2050), the thermal time ($\theta_{1}\;\left(50\right))$ in the English model (HADGEM2-ES) is accumulated in 16 days and 21 h, in the French model (CNRMCM5) in 16.5 days (Figure 5a). In the German model (MPI-ESM-LR) the thermal time ($\theta_{1}\;\left(50\right))$ is accumulated 4.5 days earlier than the current scenario, while in the American model (GFDL-CM3) the thermal time ($\theta_{1}\;\left(50\right))$ accumulated 15.5 days earlier than the current scenario (Figure 5a).

For the distant future (2070) under the conservative scenario (RCP4.5 Watts/m^2^), the French model (CNRMCM5) predicts an increase of 2.6 °C for the month of March; in the American model (GFDL-CM3) this is 3.4 °C, in the English model (HADGEM2-ES), 3.3 °C, and in the German model (MPI-ESM-LR) the predicted increase is 2.8 °C.

With a 2.6 °C increase under the French model (CNRMCM5) the thermal time ($\theta_{1}\;\left(50\right))$ accumulates 4.5 days earlier than the current scenario (Figure 5b). The German model (MPI-ESM-LR) accumulates the thermal time 4.2 days earlier than the current scenario. In the English model (HADGEM2-ES) it accumulates in 15 days and 5 h. With the 3.4 °C temperature increase predicted by the American model (GFDL-CM3), the thermal time ($\theta_{1}\;\left(50\right))$ accumulates 5.5 days earlier than the current scenario (Figure 5b).

### 2.5. Potential Distribution under Climate Change Scenarios

The prediction models generated had area under the curve (AUC) values above 0.95, which indicates a good fit of the models to the prediction of the area with the highest probability of finding the species, as a function of the temperature thresholds. According to the CCSM4 model, the distribution of *S. macrophylla* populations will tend to increase in the future (Figure 6). According to the jackknife method implemented in MaxEnt, the variables that most contributed to the current scenario were: precipitation during the driest month (54.4%), minimum temperature of the coldest month (11.1%), temperature seasonality (6.4%) and annual temperature range (6.2%).

In the near future (2050), with a representative CO_2_ concentration pathway (RCP) of 2.6 and 8.5 Watts/m^2^, an increase of locations with optimal conditions in Mexico is expected for *S. macrophylla* (18.3 and 14.4%, respectively), compared to the current distribution (Table 2). Under the RCP2.6 scenario, a stronger increase of locations with optimal climate for the species is expected. The states with the greatest increase of potential area for the species distribution were: San Luis Potosí (315.5%), Michoacán (109.2%), Querétaro (98.9%), Tabasco (48.3%) and Campeche (46.4%). In addition, optimal conditions for the species distribution are predicted to be present in Colima, Jalisco, Tamaulipas and Guerrero. In contrast, the states where it is predicted the species will not be present due to changes in the temperature are Hidalgo (22.2%) and Chiapas (5.3%). According to the jackknife method implemented in MaxEnt, the variables that most contributed to the model were: precipitation during the driest month (57.1%), annual temperature range (13.3%), altitude (7.4%) and minimum temperature of the coldest month (4.7%).

Under the RCP8.5 scenario, the states where the largest increase in optimal habitat for the species potential distribution were San Luis Potosí (140.6%), Tabasco (55.4%), Campeche (39.6%) and Yucatán (19.4%). Contrarily, the states where it is predicted the species will not be present due to changes in the temperature were: Querétaro (100%), Hidalgo (27%), Michoacán (15.6%), Chiapas (10.8%) and Puebla (3.5%). Therefore, under this scenario there was a 3.6% decrease in the optimal habitat for this species compared to the RCP2.6 scenario. According to the jackknife method implemented in MaxEnt, the variables that most contributed to the model were: precipitation during the driest month (57.6%), annual temperature range (12.2%), and minimum temperature during the coldest month (7.8%) and altitude (5.6%).

In the distant future (2070) there will be an increase of the surface in Mexico, with optimal conditions for the species distribution of 17% under the RCP2.6 and 12.3% under the RCP8.5 scenario compared to current distribution (Table 2). Under the RCP2.6 scenario, San Luis Potosí (319%), Tabasco (50.6%), Michoacán (42.2%), Campeche (40.6%) and Oaxaca (24.8%) are expected to increase in optimal habitat for the species potential distribution compared to the current distribution. In addition, optimal conditions for the distribution of the species are expected to appear in Colima, Jalisco and Tamaulipas. Meanwhile, the states where the surface with optimal conditions will decrease are Querétaro (99.8%), Hidalgo (23.6%) and Chiapas (5.1%). According to the jackknife method implemented in MaxEnt, the variables that most contributed to the model were: precipitation during the driest month (58.5%), annual temperature range (13.4%), altitude (6.7%) and minimum temperature during the coldest month (4.2%).

Under the RCP8.5 scenario, the states that will most increase in habitat suitable for the species distribution are: San Luis Potosí (202.6%), Campeche (30.2%), Tabasco (29%), Yucatán (20.8%) and Veracruz (17.2%). In addition, Baja California Sur is predicted to have optimal conditions for the presence of the species. On the contrary, in other states the surface with optimal conditions for the species distribution will decrease including Querétaro (100%), Michoacán (45.9%), Chiapas (9.4%) and Hidalgo (7%). However, this scenario presents a decrease in potential distribution area of 4.1% compared to the RCP2.6 scenario. According to the jackknife method implemented in MaxEnt, the variables that most contributed to the model were: precipitation during the driest month (56%), annual temperature range (10.1%), altitude (8.8%) and isothermality (5.5%).

## 3. Discussion

### 3.1. Germination

The seed germination of *S. macrophylla* was influenced by temperature. The highest germination percentage was recorded in the range of 25–35 °C, which coincides with Sol-Sánchez et al. [30], who obtained 100% germination in seeds of this species at 30 °C. The highest germination percentages occurred below the optimal temperature (sub-optimal range), which coincides with results reported by Sampayo-Maldonado et al. [34] for *Cedrela odorata*. Calzada-López et al. [32] mentioned that the temperature at which the germination percentage is the highest does not always coincide with the temperature at which the fastest germination speed occurs (optimal temperature). Thus, the wide range of temperatures at which the best germination percentage occurs, according to Casillas-Álvarez et al. [35], may constitute an adaptive mechanism when facing climate change, which benefits the survival and distribution of the species.

The lowest germination percentages occurred at temperatures below 20 °C, though the germination percentages could have been higher if they had been allowed more time, but the experiment was closed in 100 days. According to Caroca et al. [36], low temperatures reduce the metabolic rates of the germination processes. At temperatures of 5 and 10 °C, no germination was recorded after 100 days. In addition, Magnitisky and Plaza [26] mentioned that recalcitrant seeds of tropical trees are sensitive to low temperatures and lose viability when exposed to temperatures below 15 °C.

At temperatures between 35 and 45 °C, the time required for germination of 50% of the seed lot decreased; this coincides with Quinto et al. [29], who mentioned that exposing seeds of *S. macrophylla* to high temperatures (>28 °C) yielded higher germination rates. According to Rajjou et al. [37], temperature has an effect on the enzymes that regulate the speed of the biochemical reactions that occur within the seeds after rehydration. The fastest germination speed occurred at 35 ± 2 °C, which coincides with results reported for *C. odorata* [34].

### 3.2. Cardinal Temperatures

In order to estimate the cardinal temperatures with higher precision, Andreucci et al. [38] recommend testing a wide range of temperatures, which was considered in this experiment with a temperature range from 5 to 45 °C. *S. macrophylla* seeds presented a range of cardinal temperatures from 12.8 ± 2.4 (Tb) to 52.7 ± 2.2 °C (Tc) which highlights its sensitivity to low temperatures, a product of the physiology of the seeds, making this a potential adaptive strategy to deal with temperature increases under climate change scenarios.

The optimal temperature was 37.3 ± 1.3 °C and is directly related with a faster germination speed, which is very similar to *C. odorata* (38 ± 1.6 °C), that share the same environmental niche [34]. This coincides with Daibes et al. [39], since they mention that the optimal temperature for tropical trees is between 20 and 40 °C. According to Gilbertson et al. [40]; Dürr et al. [41] and Caroca et al. [36], this is the temperature at which germination is fastest and a seedling is obtained in the least amount of time; which could be an advantage of the species to compete in high density forests. In addition, Grey et al. [42] indicated that the germination speed increases as temperature increases, up to a point at which the temperature increase inhibits the metabolic processes of germination and there is a change in the slope where the optimal temperature is found.

The maximum temperature for germination in this species was 52.7 ± 2.2 °C, Daibes et al. [39] found that the maximum temperature for germination in tropical tree seeds was greater than 40 °C, for example species such as *Astronium lecointel*, *Parkia nitida*, and *Schizolobium amazonicum* have more than 70% germination at 40 °C.

When the temperature increases above the optimal temperature, seed germination decreases until reaching the maximum temperature (Tc), which is the upper limit at which the seed can germinate. This is due to the denaturation of proteins, which affects the cell membrane and can lead to the death of the embryo [43]. However, there are several species that can germinate at higher temperatures, Cóbar-Carranza et al. [44] report germination rates for *Pinus contorta* above 90% at temperatures between 60 and 80 °C, and they remain viable after exposure to 120 °C.

The base temperature was 12.8 ± 2.4 °C. The base temperature for tropical tree seeds is in the range of 6 to 12 °C [39]. According to Calzada-López et al. [32], low temperatures reduce seed metabolism and protein synthesis; therefore, the process of germination requires more time to accumulate heat. Andreucci et al. [38], mention that the base temperature accurately predicts the dates of the stages of phenological development of a species. Below the base temperature, the metabolic processes stop and phenological development ceases [33].

### 3.3. Thermal Time

The seeds ripen from February to March and disperse from March to April. Each phenological phase occurs when the necessary temperature is accumulated to carry out the physiological processes. Parra-Coronado et al. [33] reported that the accumulated temperature (°Cd) is the main factor influencing phenological variation, which is therefore used to include the effects of temperature and describe the effects of phenological phases with the development of agroclimatic models. According to Parmoon et al. [45] and Stenzel-Colauto et al. [46], temperature is the most important bioclimatic factor for the regulation of germination processes. Furthermore, it defines the beginning of germination, which maximizes establishment and survival during the regeneration of the understory [47].

The study of thermal requirements is important because temperature determines the metabolism for plant development and has significant effects on the initiation, percentage, and rate of germination [48]. This is the first time that to our knowledge the thermal time has been reported for *S. macrophylla*. In this study, the thermal time for *S. macrophylla* $\left(\theta_{1}\left(50\right)\right)$ in the sub-optimal temperature range was 189.091 ± 4.239 °Cd, which is higher than that, reported for *Cedrela fissilis* of 157 ± 20 °Cd [39], and much more than the thermal time for *C. odorata* of 132.74 ± 2.6 °Cd [34]. According to Parra-Coronado et al. [33] and Normand and Léchaudel [49], the late-successional tropical trees require shorter times to reach different phenological stages than the climax trees. In that sense, Norden et al. [50] mentioned that the thermal time for germination in tropical trees, being so varied, ensures the progressive emergence in the successional status to guarantee the establishment and survival of the seedlings. Moreover, Asseng et al. [51] reported that temperature accelerates phenological development as an adaptive strategy of the species to compete.

### 3.4. Germination under Climate Change Scenarios

For the intermediate future (2050) and distant future (2070), the French (CNRMCM5), American (GFDL-CM3), English (HADGEM2-ES) and German (MPI-ESM-LR) models predict an increase in temperature due to the high emissions of greenhouse gases, which will lead to the thermal sum (°Cd) accumulating in less time for *S. macrophylla*. This coincides with previous report for *C. odorata* [34]. According to Rajjou et al. [37] optimal temperatures tend to be higher for tropical trees, were germination speed increases, and faster germination ensures the establishment of individuals to compete for space [47].

For Mexico, the climate scenarios predict an increase in temperature, which will impact the processes of seed germination, leading to changes in the species distribution. According to Sánchez-Rendón et al. [52], climate change will provoke extreme drought and flood events, which will impact the functioning of tropical ecosystems, affecting the flowering, fruiting, dispersion, germination, and establishment periods of forest species. This will affect species distribution and abundance, altering the sustainability of a region [53]. However, it has been predicted that tropical species should be able to adapt to the new conditions of scarce precipitation by altitudinal and latitudinal migrations [52].

### 3.5. Potential Distribution under Climate Change Scenarios

In the present study, temperature was considered as the main prediction criterion of the potential distribution of *S. macrophylla* under climate change scenarios. It has been recognized by different authors that temperature is one of the most important bioclimatic elements to determine the response of seeds to changing environmental conditions [54] and especially the distribution of forest species [55]. Likewise, it has been mentioned that most of the physiological processes involved in the growth and development processes of trees are strongly influenced by temperature, which affects the hormonal mechanisms involved in flowering and fruiting [56], which favors the dispersion and therefore the distribution of the species.

The modeling of the mahogany tree potential distribution, according to the NCAR (CCSM4) model, there is an expected increase of 12.3 to 18.3% in the habitat with optimal temperature for *S. macrophylla* for the years 2050 and 2070 with RCP2.6 and RCP8.5. This contrasts with findings by Garza-López et al. [14], who projected that by 2030 there would be a loss of 60% of the climate habitat for *S. macrophylla* in the Yucatan Peninsula, Guatemala, Belize and eastern Honduras. Which was based on averaging 18 climatic scenarios with concentrations of RCP6.0. In this study it was decided to use the CCSM4 model, since according to Montero-Martínez et al. [57] this model has a more adequate resolution for regional impact studies. However, the projections of this model had an excellent fit as a function of the area under the curve (AUC > 0.95) [58].

According to the CCSM4 model, the states where climate habitat for *S. macrophylla* will increase the most are San Luis Potosí, Tabasco, Campeche, and Yucatán. In addition to an increase in temperature, the states of Colima, Tamaulipas and Baja California Sur have optimal temperature for the species distribution. Meanwhile, Querétaro, Chiapas and Hidalgo will decrease their areas with suitable habitat for the species. Previous studies have found discrepancies regarding the populations of *S. macrophylla* in the Yucatán Peninsula. Navarro-Martínez et al. [9] reported the populations are stable and abundant and community forest management is recommended. On the contrary, Garza-López et al. [14] reported the suitable climate for the distribution of the species has practically disappeared from the states of the Yucatán Peninsula, the most extreme case being Quintana Roo. It is therefore recommended to collect seeds in all the regions of the current distribution as a strategy to conserve the representative genetic diversity of populations threatened by climate change.

In this study, we obtained data on the cardinal temperatures and thermal time for *S. macrophylla*, which constitute a contribution to the ecophysiology of the species and are required parameters for propagation and reforestation programs, and germplasm conservation of endangered species. As such, the study of cardinal temperatures will be crucial for developing strategies to deal with climate change in tropical forests. Mexico represents the northern limit of the distribution of *S. macrophylla* in the American continent and the results presented in this study are based on the northernmost populations of the country, which is why they are considered representative of the germinative behavior of this species in Mexico, as well as the effect that climatic change can be predicted. According to Jaganathan and Biddick [59], the microclimate of the maternal environment in which seeds mature is already being affected by climate change. They mention that fruits exposed to higher temperatures develop seeds with lower moisture content, which increases the probability of cases of viviparity or loss of viability in seeds sensitive to desiccation, which would limit their dispersal. Therefore, the physiology of the seeds is already being affected from the maternal environment by the increase in temperature. To our knowledge this is the first report of cardinal temperatures for germination in species with intermediate seeds of forestry interest. The results obtained will allow better understanding of the current pattern of distribution of the species, as well as possible changes in their distribution under different climate scenarios.

All biological processes are regulated by temperature as one of the most important environmental factors affecting tree development and growth [60]. However, there are few studies that address and describe the effects of temperature on seed germination in important trees to forestry. Therefore, the study of cardinal temperatures for germination will be strategic for the development of programs for the conservation and restoration of the habitat of *S. macrophylla*, which propagates mainly by seeds, and this guarantees the genetic variability of its populations [61].

## 4. Materials and Methods

### 4.1. Thermal Thresholds for Seed Germination

#### 4.1.1. Seed Lot Details

Mature *S. macrophylla* capsules were collected in March 2018 (5 kg, 9250 seeds) from 15 trees as a representative sample from a population in Zozocolco de Hidalgo, in the state of Veracruz (649363.00 E and 2224037.00 N; 243 m.a.s.l.), in the Totonacapan region [62]. The climate is warm and humid (Af, according to García, [63]), with a mean annual precipitation of 2233 mm and average temperature of 23.4 °C, available from world climate data (http://es.climate-data.org/ (accessed on 11 June 2021)). Figure 7 shows monthly average temperature and precipitation values.

All capsules opened naturally in less than one week. The seeds were manually separated from plant debris and stored in paper bags (15 °C and 55–60% humidity) for five days, then germination tests were carried out in the Plant Physiology Laboratory of the Biotechnology and Prototype Research Unit (Unidad de Investigación de Biotecnología and Prototipos-UBIPRO) at the Faculty of Higher Education, Iztacala (Facultad de Estudios Superiores Iztacala-FESI), of the National Autonomous University of Mexico (Universidad Nacional Autónoma de México-UNAM), in Tlalnepantla, Mexico State. Following Sampayo-Maldonado et al. [64], the seeds were first washed with soap, then rinsed under running water for ten minutes, and finally rinsed three times with distilled water. They were soaked in Captan^®^ (1 g L^−1^) for 60 min, before sowing (allowing them to dry for 5 min).

#### 4.1.2. Effect of Temperature on Germination

Seeds were sown under sterile conditions in a laminar flow cabinet (Novatech, Mod. CF-13, Mexico City, Mexico). A total of 10 seeds were placed in each Petri dish with agar (10 g L^−1^), with ten replicates. The dishes were sealed with parafilm and labeled and placed in germination chambers at the constant temperature range 5–45 ± 2 °C (in five degrees intervals), under a photoperiod of 12 h light, 12 h darkness, using halogen lamps at a light intensity of 28.05 µmol m^−2^ s^−1^ (Quantum Meter Apogee Mod. QMSW-SS, Logan, UT, USA). Seeds were sown on 2 July 2018 (stored at 15 °C and 55–60% humidity, for three weeks before use). Seeds were monitored daily for the following 100 days in order to determine the proportion of germinated seeds. Following ISTA [65], Parmoon et al. [45] and Peng et al. [66], a seed was considered germinated when the radicle measured ≥2 mm, measured with a stainless-steel Vernier ruler (Truper, Mexico City, Mexico).

#### 4.1.3. Data Analyses

The number of germinated seeds in each Petri dish of the ten repetitions was recorded, to obtain averages for each treatment. The proportion of germinated seeds was calculated using the following Equation (1), where the empty seeds were eliminated [64]:(1)$$G\left(\%\right)=\frac{n}{N}*100$$
where: *n* is the number of seeds germinated and *N* is the total number of viable seeds.

The total number of days between imbibition and 50% of the total germination was recorded. Following Ordoñez-Salanueva et al. [31] a sigmoidal curve was fitted to the accumulated germination percentage, allowing the median germination time to be determined by interpolation.

This is an estimation of the number of seeds germinated per day according to the following Equation (2) [32]:(2)$$GS=\frac{G_{1}}{N_{1}}+\frac{G_{2}}{N_{2}}+\cdots +\frac{G_{i}}{N_{i}}+\cdots +\frac{G_{n}}{N_{n}}=\overset{n}{\underset{i=1}{\sum }}\frac{G_{i}}{N_{i}}$$
where: $G_{i}\;$is the number of germinated seeds in time *i* and $N_{i}\;$is the time *i* (number of days) since the Petri dishes with seeds were placed in the germination chambers.

We used a completely randomized design. The data did not fulfill the assumptions of normality, thus prior to the analysis of variance (ANOVA), percentage values (Y) were transformed using the arcsine square root function (with the original value expressed as a proportion) (T = arcsine (√Y)) [67,68]. The ANOVAs were carried out in SAS statistical software (Cary, NC, USA) [69], and the Tukey test was performed for multiple comparisons to determine significant (*p* ≤ 0.05) differences between treatments.

The base temperature (Tb) represents the temperature in degrees Celsius below which germination does not occur. Tb is the intercept point of a positive linear regression obtained when plotting the inverse time to germination as a function of temperature and it determines the sub-optimal range of temperatures. Following Ellis et al. [70] a linear regression was calculated to obtain the parameters for each germination percentage. The mean value of x-intercept (β0) was calculated and used to generate a second linear regression for each percentile. The mean β0 was the base temperature.

Following Hardegree [71], the ceiling temperature (Tc) is the maximum temperature in degrees Celsius above which germination does not occur. Tc is the intercept point of a negative linear regression obtained when plotting the inverse time to germination as a function of temperature. The correlation determines the supra-optimal range of temperatures, which was used to generate a linear regression to obtain the parameters for each germination percentage. The mean value of the x-intercept (β0) was obtained and used to do a second linear regression. The mean β0, again calculated from the second regression, represents the maximum temperature.

The optimal temperature, To, represents the optimum germination temperature in degrees Celsius at which the germination is the fastest. It was obtained by equating the regression line equations of sub and supra-optimal temperature range, the intersection value was the estimated To [71].

The inverse of the slope of the regression lines for each percentile were calculated separately to estimate the thermal time (θ, °Cd) in the sub-optimal (θ_1_) and supra-optimal (θ_2_) temperature range. The thermal time is the heat units the seeds need to accumulate for a given percentile (*g*) to complete germination in chronological time [72]. Percentage data were transformed using Probit Analysis in Genstat (version 11.1.0.1504, International Ltd., Hemel Hempstead, Herts, UK). Linear regression for the sub- and supra-optimal temperature range was used to express probit (*G*) as a function of θ.

For the germination process in the sub-optimal temperature range, the inverse of the slope of the regression lines was calculated for each percentile and the germination percentage data were transformed into probits. The sub-optimal thermal time was calculated following the Equation (3) below [73]:(3)$$Probit\;\left(g\right)=K+\left[\theta_{1}/\sigma \right]$$
where: *K* is an intercept constant when the germination progress is zero and σ is the standard deviation of the seed population response at thermal time $\theta_{1}$. This equation was used for the 50th percentile (*g*), thus, we determined the time required for germination of 50% $\left(\theta_{1}\left(50\right)\right)$ of the population.

The following Equation (4) describes the accumulated temperature in a time the seeds required to the germination process in the supra-optimal range of temperatures [73]:(4)$$Probit\;\left(g\right)=Ks+\left(T+\theta_{2}/t\left(g\right)\right)/\sigma$$
where: Ks is an intercept constant when germination is zero; $\left(T+\theta_{2}/t\left(g\right)\right)$ is the maximum temperature (Tc) and σ is the standard deviation of the seed population response. This equation was used for the 50th percentile (*g*), thus, the thermal time required for germination of 50% $\left(\theta_{2}\left(50\right)\right)\;$of the population was calculated.

#### 4.1.4. Prediction of Germination under Climate Change Scenarios

Following Fernández-Eguiarte et al. [74], we used the projected mean temperature layers proposed by the Global Circulation Model of the French model (CNRMCM5), the American model (GFDL-CM3), the English model (HADGEM2-ES) and the German model (MPI-ESM-LR), available in the Digital Climate Atlas for Mexico (http://atlasclimatico.unam.mx/AECC/servmapas/ (accessed on 27 June 2021)). According to Manzanilla-Quiñonez et al. [75], these models were generated from Phase 5 Regional Models of the Coupled Model Intercomparison Project of the Intergovernmental Panel on Climate Change (IPCC), projected for an intermediate future horizon (2045–2069) and distant future horizon (2075–2099). For this study, a representative concentration pathway (RCP) of 4.5 Watts/m^2^ (constant CO_2_ emissions), classified as a conservative scenario, will be used for the prediction. The temporal resolution was monthly and the spatial resolution was 30 × 30 s (approximately 926 × 926 m).

The seed collection region was located on the map and the average temperature data was obtained for each Global Circulation Model (GCM), Representative Concentration Pathway (RCP), and future projection. Following Cámara-Cabrales and Kelty [23], the average temperatures were projected for the current climate in March and for future scenarios in the same month, which is when seed dispersal begins.

Each scenario was used to predict the time over which the seeds of *S. macrophylla* accumulate the thermal time necessary for germination of 50%. Following Flores-Magdaleno et al. [76], the analysis was done using the mean monthly temperature method, using the following Equation (5) from Orrù et al. [77]:(5)$$Thermal\;sum\;\left({}^{\circ}\mathrm{Cd}\right)=\;\left(Env{\;T}_{m}-T_{b}\right)t_{m}$$
where: *Thermal sum* is the number of degree days accumulated, T_b_ is the base temperature for germination, T_m_ is the mean temperature for the month (m) and t_m_ is the number of days in the month (m).

### 4.2. Potential Distribution under Climate Change Scenarios

The distribution models of *S. macrophylla* were built in MaxEnt with the presence data of the specie available in georeferenced databases for the country and the climatic variables of BioClim were used.

Data acquisition: We constructed a database of georeferenced specimens collected in the country using information contained in the Global Biodiversity Information Facility (GBIF) platform from specimens deposited in herbaria around the world (https://www.gbif.org/ (accessed on 5 June 2021)). The database was cleaned to eliminate incomplete data or references located in urban or agricultural zones.

Climate variables: We used 19 climate variables at a spatial resolution of 0.3 arc minutes obtained from the BioClim database (http://www.worldclim.org/ (accessed on 6 June 2021)) for the period from 1970 to 2000 for the current distribution [78], at a resolution of 1 km^2^ per pixel [79]. The layers in vector format of land use and vegetation were taken from the Mexican National Commission for the Knowledge and Use of Biodiversity [80].

Current distribution: To obtain the area of the optimal climate habitat under the current distribution, we used version 2.0 of BioClim with data from the period of 1970 to 2000. Species distribution models (SDM) were built with MaxEnt (software version 3.4.1^®^) [81]. This process is also known as ecological niche modeling [82,83] which, since we are including only climate variables, could be considered climate niche modeling [14]. The jackknife method was implemented in MaxEnt, which indicates the relative contribution of each climate variable to the model. The program was run again using the climate variables that most contributed to the model to generate maps. The spatial distribution of the optimal climate habitat was obtained from the tool ArcMap 9.3^®^, obtaining the number of pixels and transforming to km^2^. Finally, we generated distribution maps of the species in the contemporary climate with a probability greater than 20%.

Future distribution: To obtain the area with optimal climate habitat for the future distribution we used version 1.4 of BioClim. Species distribution models (SDM) were built with MaxEnt (software version 3.4.1^®^) [81]. We downloaded the climate layers from the General Circulation Model (GCM) for the United States National Climate and Atmospheric Research Center (NCAR), the CCSM4 model, which were generated using Phase 5 Regional Models of the Coupled Model Intercomparison Project of the Intergovernmental Panel on Climate Change (IPCC), projected for an intermediate future horizon, the year 2050 (averaging from 2041–2060), and for 2070 as a distant future horizon (averaging for 2061–2080) using representative concentration pathways (RCP) of 2.6 and 8.5 Watts/m^2^ (lowest and highest CO_2_ emissions) [74]. The jackknife method was implemented in MaxEnt, which indicated the relative contribution of each climate variable to the model. Using the climate variables that contributed most to the model, the program was re-run to generate maps. The spatial distribution of optimal climate habitat was obtained using the ArcMap 9.3^®^ tool, obtaining the number of pixels and transforming to km^2^. Finally, we generated distribution maps of the species under the future climate scenarios, with a distribution probability greater than 20%.

Validation of distribution models: For model validation, we followed the recommendations of Peterson and Soberón [84]. The test was carried out using 30% of the data, which were chosen at random from all the locations where the species was present. The goodness of fit of the model predictions was evaluated using the area under the curve (AUC) of the receptor operative characteristics (ROC).

## 5. Conclusions

The germination of *S. macrophylla* was influenced by temperature. The cardinal temperatures for germination spanned a wide range. The highest speed and percentage of germination occurred at the optimum temperature (37.3 ± 1.3 °C). The thermal time $\left(\theta_{1}\left(50\right)\right)$ was estimated to determine the heat units the population needs to accumulate for seed germination. The climate change models predicted an increase in temperature, which will shorten the amount of time needed for the seeds to accumulate the temperature necessary for germination. In general, the increase in temperature will have a positive effect on germination rate. Furthermore, the CCSM4 model predicts an increase of 12.3 to 18.3% in the habitat with optimal temperature for the distribution of the species. It is recommended to locate populations that will no longer have optimal environmental conditions for the species and collect seeds to conserve and regenerate genotypes for habitat restoration programs. Moreover, it will be necessary to carry out provenance trials and assisted migration activities in future distribution areas according to these models to help generate sustainable management strategies for the long-term conservation of the species.

## Figures and Tables

**Figure 1 plants-10-02377-f001:**
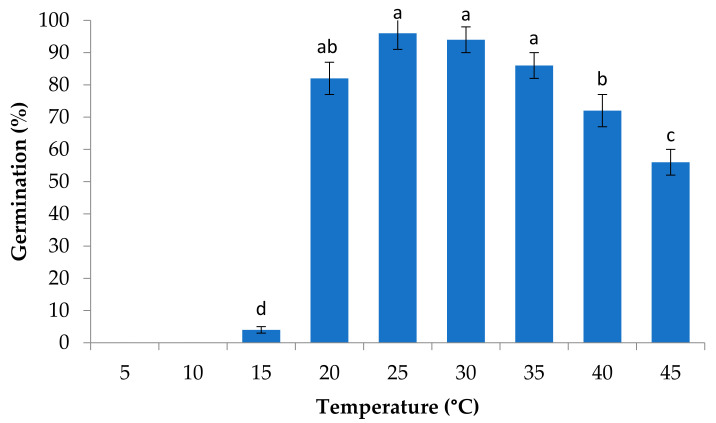
Germination percentages for each temperature treatment. Error bars show standard deviation. Means that share a letter are not significantly different (*p* ≤ 0.05).

**Figure 2 plants-10-02377-f002:**
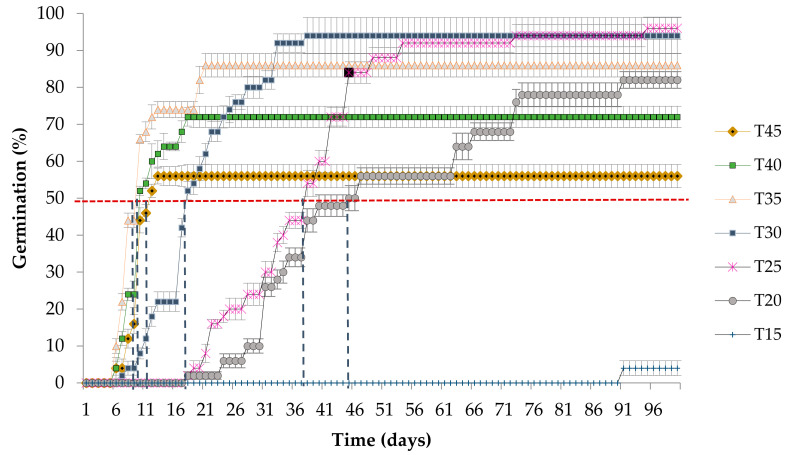
Cumulative seed germination percentages and time required for 50% germination at each temperature treatment (dashed vertical lines in blue). The error bars show standard deviation.

**Figure 3 plants-10-02377-f003:**
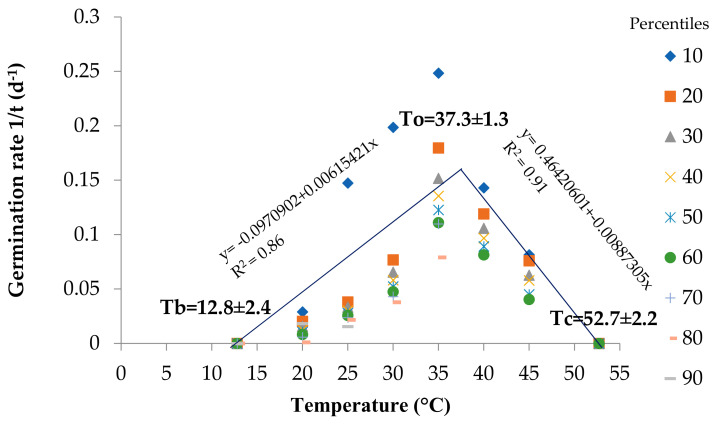
Germination rate in percentiles (inverse of germination speed) for each temperature describing the cardinal temperatures of *S. macrophilla*, (Tb: Base temp; To: Optimal temp; Tc: Maximum temperature). Regressions are based on all percentiles.

**Figure 4 plants-10-02377-f004:**
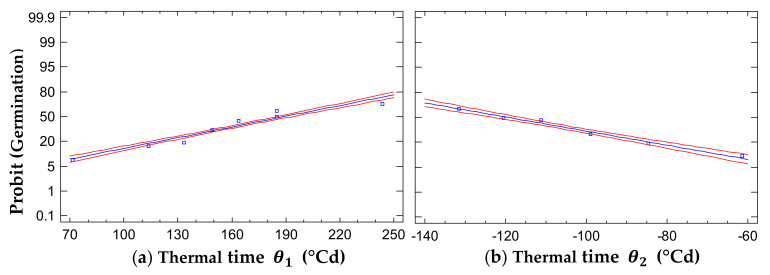
Germination in a Probit scale as a function of thermal time for: (**a**) sub-optimal range, $\;\theta_{1}$, (**b**) supra-optimal range,$\;\theta_{2}$, of temperatures. The red lines are confidence intervals of germination. The blue line is the estimated data. The points are the average of the experimental data.

**Figure 5 plants-10-02377-f005:**
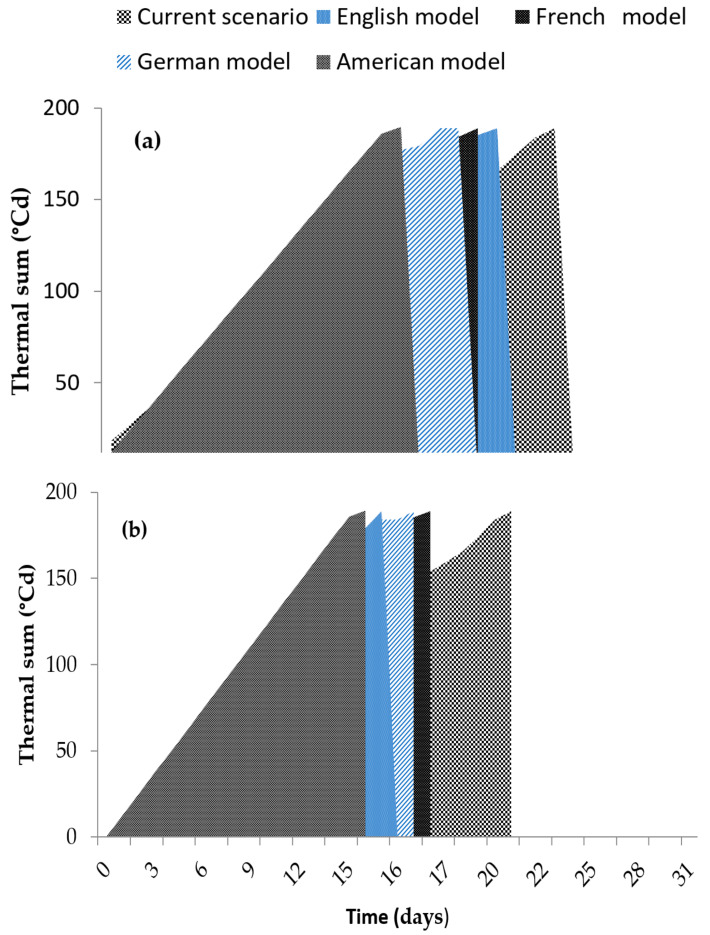
Time in which seeds accumulate the thermal sum (°Cd) during the month of March. (**a**): For the intermediate future (2050), (**b**): and for the distant future (2070). Under a conservative scenario (RCP4.5 Watts/m^2^).

**Figure 6 plants-10-02377-f006:**
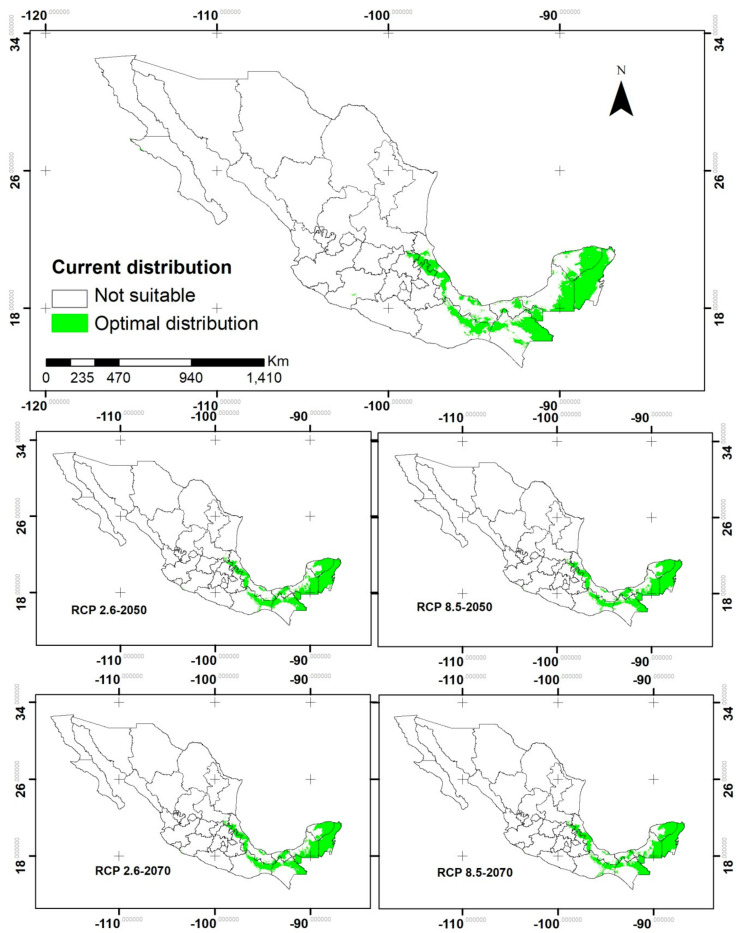
Potential distribution of *S. macrophylla*, currently and projected for 2050 and 2070 under the RCP2.6 and RCP8.5 in the CCSM4 model.

**Figure 7 plants-10-02377-f007:**
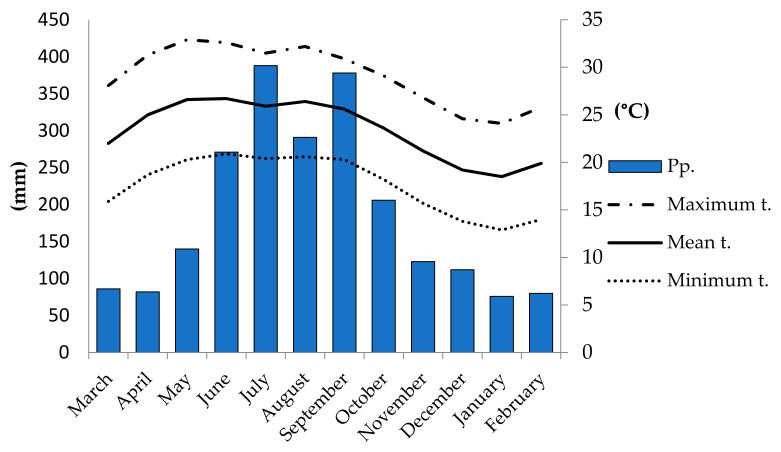
Climate data from the Zozocolco de Hidalgo, Mexico. Where *S. macrophylla* seeds were collected. (t: temperature). Bars indicate precipitation, while lines indicate maximum (dashed and dotted line), mean (solid line), and minimum temperature (dotted line). Historic monthly means from 1981–2010.

**Table 1 plants-10-02377-t001:** Thermal time parameters estimated by the Probit analysis.

Parameters	Sub-Optimal	Supra-Optimal
R^2^	92.40	96.67
K	−2.098 ± 0.079	−2.615 ± 0.160
σ	0.011 ± 0.0004	−0.021 ± 0.0014
$\theta \;\left(50\right)$	189.091 ± 4.239	119.613 ± 3.721

The valor represent mean ± standard deviation.

**Table 2 plants-10-02377-t002:** Current and predicted area (km^2^) by state where *S. macrophylla* populations could be present based on climate change scenarios according to the CCSM4 model.

State	Current (km^2^)	2050 (km^2^)		2070 (km^2^)	
		RCP2.6	RCP8.5	RCP2.6	RCP8.5
Baja California Sur	0	0	0	0	0.72
Campeche	21,412.89	31,365.56	29,896.38	30,114.68	27,863.44
Chiapas	27,479.77	26,003.84	24,489.62	26,061.28	24,890.59
Colima	0	50.02	121.63	38.45	65.12
Guerrero	0	0.56	0	0	0
Hidalgo	2550.49	1983.95	1860.27	1948.53	2371.85
Jalisco	0	9.26	1.64	3.30	0
Michoacan	194.55	407.16	164.15	276.73	105.23
Oaxaca	14,827.20	18,140.89	16,513.94	18,507.16	16,201.66
Puebla	3960.20	3990.64	3820.09	4108.47	4078.78
Queretaro	0.96	1.92	0	0.0017	0
Quintana Roo	34,786.42	40,940.55	41,018.32	41,036.26	40,907.89
San Luis Potosi	334.73	1391.83	805.46	1402.82	1013.18
Tabasco	8097.83	12,011.93	12,588.40	12,203.20	10,453.17
Tamaulipas	0	1.06	0.53	0.53	0.53
Veracruz	18,636.29	19,245.85	18,644.78	19,441.13	18,957.78
Yucatan	20,412.74	25,139.63	24,373.32	23,628.74	24,669.52
Total	152,694.13	180,684.73	174,298.59	178,771.32	171,579.54

## Data Availability

Not applicable.

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
