# Peer review of "Thermal Niche for Seed Germination and Species Distribution Modelling of Swietenia macrophylla King (Mahogany) under Climate Change Scenarios"

_plants, 2021, doi:10.3390/plants10112377_

Round 1

Reviewer 1 Report

I was interested in your research, especially in the methodology for determination of optimal, base, and maximum temperatures of the seed germination. But because of some reasons, I mention below, I don’t think this manuscript is suitable for publication in this journal yet.

#1

As the authors themselves mention, using seeds collected in just one location is critical considering probable intra specific variation, that have been possibly occurred in accordance with the present weather condition.  Just ‘RECOMMENDATION to collect seeds in all the regions of the current distribution’ is not enough. I think further discussion about this problem is essential.

#2

As well known, physiological and ecological process of germination, and ultimately reproductive success, is not determined only by average temperature provided by the climate change scenarios.  I know you restricted the significance of the research to ‘thermal niche for seed germination’. But I think it’s necessary to discuss the limitation of your data more deeply and clearly.

#3

Some additional minor comments are below.

In equation (2), i should be defined.

Concerning Figure 3;

Generaly,

v=dx/dt, where v represent velocity, or speed.

In this context,

GS=dG/dd, where GS and G respectively represents Germination Speed and the number of Germination.

So dimension of GS should be [d-1], and that of inverse of GS should be[d].

I was confused by your explanation of y-axis.

In addition, legend of each plot should be presented clearly.

Author Response

REVIEWER 1

#1

As the authors themselves mention, using seeds collected in just one location is critical considering probable intra specific variation, that have been possibly occurred in accordance with the present weather condition. Just ‘RECOMMENDATION to collect seeds in all the regions of the current distribution’ is not enough. I think further discussion about this problem is essential.

The following text was added in the discussion to respond to this comment:

Mexico represents the northern limit of the distribution of S. macrophylla in the American continent and the results presented in this study are based on the northernmost populations of the country, which is why they are considered representative of the germinative behavior of this species in Mexico, as well as the effect that climatic change can be predicted. According to Jaganathan and Biddick [59], the microclimate of the maternal environment in which seeds mature is already being affected by climate change. They mention that fruits exposed to higher temperatures develop seeds with lower moisture content, which increases the probability of cases of viviparity or loss of viability in seeds sensitive to desiccation, which would limit their dispersal. Therefore, the physiology of the seeds is already being affected from the maternal environment by the increase in temperature.

#2

As well known, physiological and ecological process of germination, and ultimately reproductive success, is not determined only by average temperature provided by the climate change scenarios. I know you restricted the significance of the research to ‘thermal niche for seed germination’. But I think it’s necessary to discuss the limitation of your data more deeply and clearly.

The following text was added in the discussion to respond to this comment

In the present study, temperature was considered as the main prediction criterion of the potential distribution of S. macrophylla under climate change scenarios. It has been recognized by different authors that temperature is one of the most important bioclimatic elements to determine the response of seeds to changing environmental conditions [54] and especially the distribution of forest species [55]. Likewise, it has been mentioned that most of the physiological processes involved in the growth and development processes of trees are strongly influenced by temperature, which affects the hormonal mechanisms involved in flowering and fruiting [56], which favors the dispersion and therefore the distribution of the species.

#3

Some additional minor comments are below.

In equation (2), i should be defined.

            Done

Concerning Figure 3;

Generaly,

v=dx/dt, where v represent velocity, or speed.

            v means germination speed.

In this context,

GS=dG/dd, where GS and G respectively represents Germination Speed and the number of Germination.

So dimension of GS should be [d-1], and that of inverse of GS should be[d].

I was confused by your explanation of y-axis.

            The description of percentiles was included in the legends and figure captions.

In addition, legend of each plot should be presented clearly.

            Legends were added to each axis of the Figure.

Reviewer 2 Report

The manuscript “Thermal niche for seed germination and species distribution modelling of Swietenia macrophylla King (mahogany) under climate change scenarios” is very well prepared and organized. The presented topic is of potential audience interest. The Abstract is well balanced and presents really a brief summary of the performed study. The Introduction part introduces well to the topic under discussion and highlights the current challenges in the topic under study. The method part is very well described with all required details enabling the reconstruction of the experience. The results are well presented and discussed with the current literature. The Tables and Figures are well prepared with all details and made in high quality (easy to read and understand). The conclusion is clearly described and supported by the obtained results. Overall quality is good, therefore the present manuscript can be advised for publication in the present form.

Author Response

The reviewer has no questions and we welcome the reviewer's comments.

Reviewer 3 Report

The authors have investigated how predicted climate warming will affect the germination dynamics, and subsequently, the future distribution of Swietenia macrophylla (mahogany) in Mexico. In my estimation, the study is methodologically sound, well written and of general interest to the readers of Plants. As such, I recommend this manuscript be accepted for publication with the following minor revisions:

L24 – It isn’t clear what “its” is referring to in this sentence. I would replace with “the potential biogeographic distribution”.

L31-32 – I’m lost as to how the second part of this sentence relates to the first. Could you please clarify?

L60 – Please add “and” before “is affected by a shrinking gene pool”.

L65 – change “as they do not…” to “as they cannot…”.

L66-68 – Again, I’m a little confused how this sentence relates to what came before it.

L73 – Please add “is” before “threatening the ecosystem…”.

L97 – Change “factor” to “factors”.

L100 – Change “there are not publications reporting…” to “no prior study has reported…”.

Figure 1 caption – add “treatment” after “temperature”.

L114 – Please add “observed” after “the lowest germination percentage was…”.

L136 – This would sound better as “We observed significant differences in germination speed among treatments”.

Generally: the Tb, To, and Tc notation needs to be described earlier and explained to the reader. I was not aware of these terms and they confused me until I had read further into the manuscript.

L173 – Please add “accuracy” after 92%.

Figure 3 – the legend here needs to be explained. What do the different point types denote? Percentage germination? A legend is put to the right of the graph, but no explanation is given in the caption or labelled on the legend.

L349-350 – I would be careful here. I would add “potential adaptive strategy”.

L411 – change “tropical tree” to “tropical trees”.

General point for the discussion: see also Jaganathan & Biddick (2020). If the maternal environment has strong influences on germination dynamics, perhaps climate warming could indirectly affect seeds via the maternal environment.

Other than the above, great study and well done.

References:

Jaganathan GK, Biddick M (2020). Critical role of air and soil temperature in the development of primary and secondary physical dormancy in Albizia julibrissin (Fabaceae). Journal of Tropical Ecology 36(6): 1-7.

Author Response

Reviewer 3

L24 – It isn’t clear what “its” is referring to in this sentence. I would replace with “the potential biogeographic distribution”.

            Done

L31-32 – I’m lost as to how the second part of this sentence relates to the first. Could you please clarify?

            The wording of the sentence was corrected

L60 – Please add “and” before “is affected by a shrinking gene pool”.

            Done

L65 – change “as they do not…” to “as they cannot…”.

            Done

L66-68 – Again, I’m a little confused how this sentence relates to what came before it.

            The wording of the sentence was corrected

L73 – Please add “is” before “threatening the ecosystem…”.

            Done

L97 – Change “factor” to “factors”.

            Done

L100 – Change “there are not publications reporting…” to “no prior study has reported…”.

            Done

Figure 1 caption – add “treatment” after “temperature”.

            Done

L114 – Please add “observed” after “the lowest germination percentage was…”.

            Done

L136 – This would sound better as “We observed significant differences in germination speed among treatments”.

            Sounds better, done

Generally: the Tb, To, and Tc notation needs to be described earlier and explained to the reader. I was not aware of these terms and they confused me until I had read further into the manuscript.

            The definition of Tb, To and Tc was included in their first appearance on lines 150 to 153

L173 – Please add “accuracy” after 92%.

            Done

Figure 3 – the legend here needs to be explained. What do the different point types denote? Percentage germination? A legend is put to the right of the graph, but no explanation is given in the caption or labelled on the legend.

            The legends of the axes were included for a better understanding of the Figure.

L349-350 – I would be careful here. I would add “potential adaptive strategy”.

L411 – change “tropical tree” to “tropical trees”.

General point for the discussion: see also Jaganathan & Biddick (2020). If the maternal environment has strong influences on germination dynamics, perhaps climate warming could indirectly affect seeds via the maternal environment.

            The reference was included and the discussion was improved with this suggestion

Round 2

Reviewer 1 Report

I think the manuscript is now good enough for publication in Plants as an article.